# Risk factors for cardiovascular disease among Saudi students: Association with BMI, current smoking, level of physical activity, and dietary habits

Mohammed Shaab Alibrahim[1†], Mohamed Ahmed Said[1†*], Abdulmalek K. Bursais[1], Ibrahim I. Atta[1], Mohamed Abdelmoniem Abdelrahman[1], Hasnaa Hamdi Mohamed[2], Ahmad K. Hassan[1], Abdulrahman I. Alaqil[1], Norah S. Almudaires[1], Narjis M.A. Alamer[1], Osama Eid Aljuhani[3], Hind Omer Salem Alshaghdali[4], Amani Hamzah ALjahani[5], Zuhair A. Al Salim[6], Atyh Abdullah Hadadi[7], Najeeb Abbas Aldarushi[8], Amal Nassir Alkuraieef[5], Ghareeb O. Alshuwaier[9]

**1** Department of Physical Education, College of Education, King Faisal University Al Ahsa, Saudi Arabia, **2** College of Applied Medical Sciences, King Faisal University, Saudi Arabia, **3** Department of Physical Education, College of Sport Sciences and Physical Activity, King Saud University, Saudi Arabia, **4** Department of Curriculum and Instruction, Faculty of Education, Hail University, Saudi Arabia, **5** Department of Physical Sport Sciences, College of Sport Science and Physical Activity, Princess Nourah bint Abdulrahman University, Saudi Arabia, **6** Department of Sport Science and Physical Activity, College of Science, University of Hafer Al Batin, Saudi Arabia, **7** Department of Sports Science, College of Education, Taif University, Saudi Arabia, **8** Department of Sports Management, College of Sports Science, Jeddah University, Saudi Arabia, **9** Department of Exercise Physiology, College of Sport Sciences and Physical Activity, King Saud University, Saudi Arabia

† These authors have contributed equally to this work and share first authorship.
* masaid@kfu.edu.sa, said.med@laposte.net, said.med1966@gmail.com

## Abstract

Cardiovascular disease (CVD) risk factors, including poor diet, lack of physical activity (PA), smoking, and obesity, are associated with unhealthy lifestyle choices and contribute significantly to the global disease burden. This study aims to investigate the prevalence of behavioral risk factors associated with overweight/obesity, PA, smoking, and eating habits among Saudi students and explores how these vary by region, age, gender, relationship status, and income. A total of 968 participants (285 males and 681 females), aged 18–50, were recruited from bachelor's, master's, doctoral, and other university programs across the Eastern, Riyadh, and Western provinces. Each participant completed the Saudi Food Frequency Questionnaire (SFFQ). The risk of CVD was assessed by summing individual risk factors related to BMI, tobacco use, PA level, glycemic load (GL) of ingested foods, and consumption of saturated fats (SF), processed meats, oily fish, nuts, and cereal fiber (CF). Participants were classified into minimal, medium, or high-risk categories based on their total scores. A high prevalence of CVD risk factors was observed, with 93.6% of participants having three or more risk factors. After adjusting for demographic variables, living in Riyadh was associated with a 12% lower probability of CVD risk. Conversely,

**Data availability statement:** All data supporting the findings of this paper are available from Zenodo at DOI: 10.5281/zenodo.14977380 (https://zenodo.org/records/14977380).

**Funding:** The Deanship of Scientific Research, Vice Presidency for Graduate Studies and Scientific Research, King Faisal University, Saudi Arabia, financed this study (KFU250925). The funders had no role in study design, data collection and analysis, decision to publish, or preparation of the manuscript.

**Competing interests:** The authors have declared that no competing interests exist.

students aged 46–50 showed a five- to nine-fold increase in CVD risk. Significant factors influencing CVD risk included BMI (Exp(β) = 11.70), smoking status (Exp(β) = 6.54), PA (Exp(β) = 7.61), SF intake (Exp(β) = 4.79), GL (Exp(β) = 7.00), CF intake (Exp(β) = 24.58), and oily fish consumption (Exp(β) = 2.99). Low CF intake and high BMI were the most prominent risk factors. Lifestyle interventions targeting overweight/obesity, physical inactivity, smoking, high GL and SF intake, and promoting CF and oily fish consumption could improve CV health among participants. Addressing these modifiable risk factors is essential for effective prevention.

## Introduction

Cardiovascular disease (CVD) encompasses various medical conditions affecting the heart and blood vessels, primarily caused by the buildup of lipid deposits in the arteries (atherosclerosis) and increased susceptibility to thrombosis. It can also result in arterial damage to vital organs, including the brain, heart, kidneys, and eyes. Key categories of CVD include coronary heart disease, stroke, transient ischemic attack, peripheral artery disease, and aortic disease [1]. According to the World Health Organization [2], CVD is the leading cause of mortality globally, responsible for approximately 17.9 million deaths in 2019, representing 32% of all deaths. Strokes and heart attacks accounted for 85% of these fatalities.

CVDs, traditionally more common in older populations, are now increasingly impacting younger demographics. In the United States, rising rates of adolescent obesity have significantly escalated healthcare expenditures and lifetime risks of CVD. Healthcare costs increased by 8.6% for overweight individuals, 18.8% for individuals with obesity, and 76% for those with morbid obesity [3]. This trend underscores the pressing necessity for early measures to prevent CVD in younger populations.

CVDs also represent a major public health challenge in Saudi Arabia. In 2012, the prevalence of heart failure was estimated at 1.2%, equating to approximately 348,000 patients in a population of 29 million [4]. Albackr et al. [5] reported that 20% of patients hospitalized for acute coronary syndrome concurrently experienced heart failure. By 2015, the prevalence of coronary artery disease in the country had reached 5.5% [6]. In 2016, 201,300 Saudi nationals were diagnosed with CVDs, including 149,600 cases of ischemic heart disease and 51,700 cases of heart failure [7].

Recent research reveals that 1.6% of the Saudi population aged 15 and older—approximately 236,815 individuals—are affected by CVDs, with a higher prevalence among males (1.9%) than females (1.4%) [8]. The prevalence of CVD is projected to rise due to increasing risk factors such as obesity, diabetes, dyslipidemia, and hypertension.

Epidemiological studies suggest that the most important behavioral risk factors for CVDs include physical inactivity [9], unhealthy diets [10], excessive alcohol consumption [11], and smoking [12]. Globalization, urbanization, and an aging population may further exacerbate the risk of CVD. Social and economic factors, mental

health issues, and genetic predispositions also contribute to CVD [2]. These behavioral risk factors often manifest as hypertension [13], hyperglycemia [14], hyperlipidemia [15], overweight, and obesity [16]. The World Health Organization emphasizes the importance of early detection of CVD to initiate therapy through counseling and medication [2].

Adopting a heart-healthy diet low in saturated fats (SF), trans fats, cholesterol, and sodium, and high in fruits, vegetables, whole grains, and lean proteins can reduce the risk of CVD [10,17]. Understanding the connections between dietary habits and cardiovascular health is crucial for developing effective dietary guidelines and interventions aimed at reducing the burden of CVD [18]. Engaging in 150–300 minutes of moderate physical activity (PA) or 75–150 minutes of vigorous PA weekly, or a combination of both, has been shown to lower the risk of developing CVD [9]. Quitting smoking [19] and moderating alcohol consumption [20]) are additional steps individuals can take to lower their risk. Pharmacological interventions are also essential for managing hypertension, diabetes, and hyperlipidemia to reduce the likelihood of heart attacks and strokes [14,15,21].

Over the past three decades, Saudi Arabia has experienced significant socioeconomic changes that have affected lifestyle and dietary habits. Traditional diets rich in fiber and vitamins have been replaced by those high in sodium, SF, and processed sugars, thereby increasing CVD risk factors [22,23]. A substantial portion of the population is overweight or obese. Alhejely et al. [24] reported that over half of their study participants were obese, with one-third having low education levels, a factor linked to higher CVD risk. Hypertension and diabetes are also widespread, with many individuals either unaware of their conditions or failing to manage them [24]. These issues are exacerbated by sedentary lifestyles and poor dietary habits [24,25]. In 2016, CVD remained the leading cause of death in Saudi Arabia, accounting for over 45% of fatalities [6].

Recognizing the severity of this challenge, the Saudi government has implemented various policies and initiatives under Vision 2030. These include national health programs promoting improved nutrition, increased PA, and stricter tobacco regulations. Alcohol consumption is excluded due to its prohibition under Islamic law, the predominant religion in Saudi Arabia. Measures to address unhealthy diets include taxing sugary and energy drinks, mandating nutritional labeling, and restricting trans fats [25]. PA campaigns aim to address the high levels of inactivity among Saudis [25]. Tobacco control measures include steep taxes on tobacco products and smoking bans in public spaces [25].

This study aims to evaluate cardiovascular risk factors linked to body composition, PA, smoking habits, and dietary behaviors among students in Saudi Arabia. The analysis assesses the variability of these risk factors based on geographical location, age, gender, marital status, and monthly income. Additionally, the effects of these risk factors on the likelihood of developing cardiovascular problems are explored.

## Methods

### Participants

This cross-sectional study encompasses the responses of 968 Saudi participants (285 men and 681 women), aged between 18 and 50, recruited between March 1, 2023, and December 31, 2023, at five universities in three provinces: East, West, and Riyadh. According to statistics provided on university websites, there are a total of 198,537 students, with 30.9% from the Eastern Province, 47.4% from the Riyadh Province, and 21.6% from the Western Province. Based on a 95% confidence level and a 5% margin of error, a minimum sample size of 384 participants is required [26], distributed as follows: 119 participants from the Eastern Province (53 male and 66 female), 182 from the Riyadh Province (89 male and 93 female), and 83 from the Western Province (39 male and 44 female). Referring to the findings of Wu et al. [27], a minimum initial sample size of 1,920 participants is recommended to yield 384 complete responses, assuming a 20% response rate due to the questionnaire's length.

The sample was randomly selected, focusing on only university attendees aged 18–50 years from various educational modalities, including regular, part-time, and distance/online learning. To ensure representation of the three primary academic domains—health and medical sciences, arts and humanities, and science and engineering—three colleges were

chosen from each university. Within these colleges, a systematic sampling approach was used to select two classes from each level of study across seven academic stages: four undergraduate levels (UG), two postgraduate levels (PG), and one distance/online learning level. All students in the selected classes were invited to participate. To achieve balanced gender representation, each level included one class of male students and one class of female students. Class sizes ranged from 5 to 20 students, resulting in a total sample size of 2,275 students, with an approximately equal gender distribution (55.5% female).

Each participant completed a culture-specific food frequency questionnaire (The Saudi Food Frequency Questionnaire, SFFQ), designed to assess dietary habits and risk factors associated with CVD among Saudis [28]. The Deanship of Scientific Research sent an email to each student containing a Google Form link. This link directed students to a page outlining the study's objectives, the estimated time required to complete the questionnaire, and instructions for participants. It was made clear that participation was entirely voluntary, with the option to withdraw at any time. All responses were anonymous, so no formal consent was needed, and participants were invited to click on the "Questionnaire" icon to get started.

Data were gathered during two intervals: March–April 2023, yielding 609 responses (26.77% response rate), and November–December 2023, yielding 679 responses (28.85% response rate). To promote participation, the university-affiliated author(s) actively engaged with students directly by visiting selected classes and indirectly by urging teachers to encourage their students to complete the questionnaire.

A total of 1,288 participants consented to participate in the study (males: n = 345; females: n = 943), resulting in a response rate of 56.62%. Data were reviewed for accurate, and incomplete or disputed questionnaires (n = 320) were removed, leaving 968 valid responses, representing 42.55% of the total surveyed participants, for analysis. Ethical approval was granted by the Research Ethics Board of King Faisal University, Saudi Arabia (KFU-REC-2022-OCT-ETHICS202).

## Measures

### Saudi Food Frequency Questionnaire (SFFQ)

We used the Saudi Food Frequency Questionnaire (SFFQ) to assess the risk of CVD and type 2 diabetes mellitus (T2DM) among Saudis [28]. The SFFQ comprises four main sections: demographics (questions 1–14), smoking habits (questions 15–18), physical activity and sedentary behavior (questions 19–25), and food consumption and eating habits (questions 26–90). Responses were employed to compute frequencies, amounts, or scores for various parameters that, when juxtaposed with benchmarks, could signify a possible risk of CVD or T2DM.

**Demographics.** Includes questions about participants' geographic region, relationship status, household income, age, and gender. Participants were asked about their health history, including non-communicable disorders such as stroke, CHD, asthma, type 2 diabetes, hyperlipidemia, disability, heart failure, angina, irregular pulse, severe palpitations, heart muscle weakness, and any other minor cardiovascular concerns. They were also asked about their height and weight. BMI was calculated by dividing the weight into kilograms by the height in square meters. The participants were classified into four weight groups based on their BMI: underweight (BMI < 18.5), normal weight (BMI 18.5–24.9), overweight (BMI 25–29.9), and obese (BMI ≥ 30) [29]. Overweight or obese patients were considered to have an elevated risk of CVD, and one point was added to their final score [30].

**Smoking.** This section looks at participants' smoking status, including whether they are smokers or non-smokers, the specific type of smoking they engage in (cigarettes, cigars, pipes, shisha), and the amount (number of cigarettes per day) or frequency (number of sessions per day and duration of each session) of smoking. Although smoking is not considered a dietary habit, it promotes the development of hypertension and CVD [31]. Hence, questions on smoking were included, as the calculation of CVD risk related to dietary behaviors needs to be corrected for smoking, where appropriate. This study considered only the binary variable of being a regular smoker. The ultimate CVD risk scores of smokers were increased by one point [32].

**Physical activity.** The third section of the questionnaire focuses on physical activity and sedentary behaviors. Instead of creating new questions, items from the Arabic version of the International Physical Activity Questionnaire (IPAQ) were incorporated. This validated version, widely recognized and freely available, measures the frequency and duration of various physical activities, including walking, moderate-intensity activities, and vigorous-intensity activities, as well as sedentary behaviors, such as time spent sitting [33]. The resulting scores categorize participants into three groups: (1) Inactive (no activity is reported or only some activity that is insufficient to meet categories 2 or 3). (2) Minimally active, reporting three or more days of at least 20 min/day of vigorous activity, five or more days of moderate-intensity activity, or walking at least 30 min/day, or five or more days of any combination of walking, moderate-intensity, or vigorous intensity activities, achieving a minimum of at least 600 metabolic equivalents (MET)-minutes/week. (3) HEPA active, reporting vigorous activity on at least three days and accumulating at least 1500 MET-minutes/week or seven or more days of any combination of walking, moderate-intensity, or vigorous intensity activities, achieving a minimum of at least 3000 MET-minutes/week [34]. Inactive participants were considered to have a high risk of CVD, and one point was added to their final scores [35].

**Food consumption and eating habits.** This section is intended to reflect both traditional Saudi eating patterns and the growing influence of Western diets. Questions about the frequency (number of times) and quantity (number of servings, spoons, milliliters, cubes, or cups) of foods ingested were asked. Foods included fruits, vegetables, cereals, dairy products, meats, nuts, processed foods, and sweetened beverages. The glycemic load (GL), SF, and cereal fiber (CF) content were measured [36].

## Dietary analysis and CVD risk

The daily GL of the foods ingested, and the amounts of CF and SF consumed were calculated based on the answers provided. The consumption of processed meat, fatty fish, and nuts was also determined. Nutritional intake was determined through the frequency of consumption of each food item in the questionnaire and the nutritional value of the corresponding portion. Participants were categorized into two (low and high) or three (low, medium, and high) risk levels according to previous studies. Energy intake was determined by summing the calories derived from carbohydrates, fats, and proteins, based on the 4:9:4 rule. Dietary information was deemed legitimate if reported energy intakes ranged from 600 to 4000 kcal/day for women and 600–4200 kcal/day for men, with less than 13 foods unaccounted for [37]. Subsequently, the CVD risk level was calculated by summing all the individual risks related to BMI, current smoking, PA, and dietary intake. The obtained score was used to categorize participants into three groups: minimal risk (scores of 0–2), medium risk (scores of 3–5), and high risk (scores ≥ 6) [38]).

The GL of the foods included in the SFFQ was used for carbohydrates. The GL values of traditional Saudi foods were determined using the food glycemic indexes reported by Al-Mssallem [39]. The mean GL values of other food items were obtained from Foster-Powell et al. [40], who developed a table that contained the GL values of 750 food categories, assessed using a standardized procedure. Three solutions were employed for Saudi items not included in the available food tables. Initially, certain Saudi food and beverages were correlated with analogous items presented in the aforementioned tables. Secondly, in instances where analogous food and beverages were absent from the aforementioned tables, Saudi recipes were used to ascertain the nutritious content, considering the recommended portion sizes indicated in the recipes. Each nutrient was thereafter recorded individually, utilizing the specified quantity and considering the portion size indicated by the participants [38].

The GL of foods was determined by multiplying the carbohydrate content of the food by its glycemic index and then multiplying the obtained value by the frequency of consumption. The latter was determined using a scale of eight possible replies, ranging from "never" to "3 or more times a day." The values assigned to each response were as follows: never = 0, 1–3 per month = 0.9, once a week = 2.8, 2–3 times/week = 7, 4–5 times/week = 12.6, once a day = 20, 2–3 times/day = 50, and > 3 times/day = 90. Finally, the total GL value of each participant's food intake was determined, and the participants

were categorized into three groups: (1) low-risk (GL for men ≤ 180/day; GL for women ≤ 117/day), (2) medium-risk (GL for men = 181–249/day; GL for women = 118–205/day), and (3) high-risk (GL for men ≥ 250/day; GL for women ≥ 206/day). The final score was increased by one point for participants classified as medium risk and two points for those classified as high risk [41].

The average fiber content of the food was considered for CF. A factor was assigned to each response for the calculation. For example, "less than once a week" had a factor of 1/7, "1–2 days a week," of 2/7, "3–4 days a week," of 4/7, and "5–7 days a week," of 1. These factors were assigned to fruit, nuts, seeds, vegetables, and legumes, and were then multiplied by the food's fiber content. The fiber estimates for each food were then added to obtain a rough estimate of fiber intake, and the participants were categorized into three groups: (1) low-risk (CF for men > 10.6 g/day; CF for women > 7.7 g/day), (2) medium-risk (CF for men = 5.3–10.6 g/day; CF for women = 7.7–3.9 g/day), and (3) high-risk (CF for men < 5.3 g/day; CF for women < 3.9 g/day). The final score was increased by one point for participants classified as medium risk and two points for those classified as high risk [42].

The procedure described below was used to determine the amount of SF and processed meat ingested per day. The SF content of foods such as eggs, cheese, milk, ice cream, yogurt, potato chips, cream, mayonnaise, cookies, chocolate, desserts, ghee, pies, butter, red meat, processed meat, and fast food was obtained from McCance and Widdowson [43] or https://www.webteb.com/nutritionfacts, which provides calories and nutritional values for the most popular Saudi foods. The amount of SF that each participant consumed was calculated by multiplying the SF content per serving by the intake frequency. The amount of processed meat consumed was determined by multiplying the serving's weight by the intake frequency. On the other hand, the number of portions per week and per day were considered for oily fish [44] and nuts [45], respectively. Women who ingested a daily SF amount ≤ 24.4 g, men who ingested a daily SF amount ≤ 30.5 g, and individuals who ingested < 50 g/day processed meat, over four servings per week of oily fish, and one or more portions of nuts daily were categorized as having a low risk of CVD. However, participants were classified as high-risk if their daily intake of SF exceeded 24.4 g/day for women and 30.5 g/day for men [46], if their consumption of processed meat exceeded 50 g/day [47], if their intake of oily fish was under than two portions/week [48], and if they did not consume nuts daily [45].

## Statistical analysis

Food intake was converted to nutrient intake using WinDiets program (Robert Gordon's University, Aberdeen, UK). To account for potential non-response bias, participants were weighted based on demographic variables to ensure representativeness of the sample. The Shapiro-Wilk test was employed to assess the distributions' normality. For continued data analysis, means were compared using the *t*-test or ANOVA for independent samples. The 95% confidence interval was determined by comparing the observed data with hypothesized data. Participants were assigned scores based on their CVD risk levels: "0" for low risk, "1" for medium risk, and "2" for high risk. The chi-square test was used to compare the prevalence of CVD risk factors between men and women. Stepwise logistic regression was applied to evaluate the impact of CVD risk factors related to overweight/obesity, smoking, PA, and dietary behaviors on the incidence of cardiovascular concerns. Demographic variables were included as covariates in the regression model to control confounding effects. Participants with a history of cardiovascular conditions received a score of 1, whereas those reporting good health were assigned a score of 0. The requisite assumptions were assessed by targeted testing using SPSS (version 26; IBM, Armonk, NY, USA), and the significance level was set at $p < 0.05$.

## Results

### Risk of non-response bias

To account for the possibility of risk of non-response bias, weights were generated for demographic variables using observed participant distributions and target population distributions [6,22–24,28]. The findings are summarized in Table 1. The estimated weights show that certain groups, particularly female students and participants from Riyadh province, were

**Table 1. Weights for demographic variables.**

| Variable | Observed Distribution (%) | Target Distribution (%) | Weight |
|---|---|---|---|
| Region | | | |
| Eastern | 570 (58.9%) | 30 | 0.51 |
| Riyadh | 177 (18.3%) | 50 | 2.73 |
| Western | 221 (22.8%) | 20 | 0.88 |
| Gender | | | |
| Male | 286 (29.5%) | 50 | 1.7 |
| Female | 682 (70.5%) | 50 | 0.71 |
| Relationship Status | | | |
| Single | 757 (78.2%) | 60 | 0.77 |
| Married | 181 (18.7%) | 30 | 1.6 |
| Divorced | 23 (2.4%) | 8 | 3.33 |
| Widowed | 7 (0.7%) | 2 | 2.86 |
| Age | | | |
| Under 20 years old | 208 (21.5%) | 25 | 1.16 |
| 20 to 25 years old | 591 (61.1%) | 40 | 0.65 |
| 26 to 30 years old | 34 (3.5%) | 15 | 4.29 |
| 31 to 35 years old | 43 (4.4%) | 10 | 2.27 |
| 36 to 40 years old | 45 (4.6%) | 5 | 1.09 |
| 41 to 45 years old | 27 (2.8%) | 3 | 1.07 |
| 46 to 50 years old | 20 (2.1%) | 2 | 0.95 |
| Income | | | |
| 3000 SAR or less | 180 (18.6%) | 20 | 1.08 |
| 3001 to 8000 SAR | 234 (24.2%) | 30 | 1.24 |
| 8001 to 13,000 SAR | 232 (24.0%) | 25 | 1.04 |
| 13,001–18,000 SAR | 143 (14.8%) | 15 | 1.01 |
| 18,001–23,000 SAR | 94 (9.7%) | 5 | 0.52 |
| More than 23,000 SAR | 85 (8.8%) | 5 | 0.57 |
| Cardiovascular Issues | | | |
| No | 932 (96.3%) | 95 | 0.99 |
| Yes | 36 (3.7%) | 5 | 1.35 |

Note: SAR, Saudi riyal.

underrepresented in our sample. As a result, these weights will be used in later statistical studies to account for demographic imbalances and reduce potential risk of non-response bias.

## Demographics

Weights and the demographic distribution of participants by area, gender, relationship status, income, and if they had cardiovascular issues were included in Table 1. Of the total participants, 570 individuals (58.9%) were from the Eastern Province, followed by 221 individuals (22.8%) from Western Province, and 177 individuals (18.3%) from Riyadh Province. Most participants were female (70.5%), aged between 20 and 25 years (61.1%), unmarried (78.2%), and earning between 3001 and 8000 Saudi riyals (SAR) (24.2%) or between 8001 and 13,000 SAR (24.0%). Thirty-six participants, accounting for 3.7%, had at least one cardiovascular condition.

The calculated weights for demographic variables indicate notable differences between the sample and the target population. Participants from the Eastern Province exhibit a disproportionate representation (weight: 0.51), as do female

students (weight: 0.71) and single individuals (weight: 0.77). In contrast, students from Riyadh Province (weight: 2.73), male participants (weight: 1.7), and divorced or widowed individuals (weights of 3.33 and 2.86, respectively) are markedly under-represented. The age range of 26–30 years exhibits significant under-representation (weight: 4.29). Educational level, especially among postgraduates, has notably high weights (10), signifying underrepresentation in this group. Income distributions align closely with desired values, indicating a precise representation in this aspect. These weights underscore the necessity for modifications in studies to rectify demographic discrepancies, hence improving the reliability and generalizability of the study's results.

**BMI, MET, food intake, and overall CVD risk**

Table 2 presents the mean and standard deviation of the BMI, MET for PA, and food intake among participants (i.e., the quantities or portions of food ingested by the participants). The t-test demonstrated that, compared to females, males had a higher BMI ($24.39 \pm 5.85$ vs. $22.85 \pm 4.81$; $p < 0.001$; Fig 1-1), ate more oily fish ($p = 0.47$) and nuts ($p = 0.049$; Fig 1-2), and burned more energy during PA ($1576.47 \pm 957.14$ vs. $1321.88 \pm 933.01$ METs; $p < 0.001$; Fig 1-3). However, women had a higher cumulative CVD risk than men ($4.71 \pm 1.64$ vs. $5.09 \pm 1.49$; $p < 0.001$; **Fig 1-2**).

Significant differences were observed in BMI ($p < 0.001$), MET-minutes per week ($p < 0.001$), CVD risk ($p = 0.003$), as well as daily consumption of SF ($p < 0.001$) and processed meat ($p < 0.001$) based on weighted region. In particular, individuals living in the West had a lower BMI and spent less energy in PA than those living in the Eastern Province ($p < 0.001$ for both) and Riyadh Province ($p = 0.039$ and $p = 0.002$, respectively; Figs 2-1 and 2-3). Moreover, students of the Eastern Province exhibited a reduced intake of SF and processed meat relative to their counterparts in the Western Province ($p < 0.001$ and $p = 0.024$, respectively) and lower SF consumption than participants from Riyadh ($p < 0.001$; Fig 2-1). Moreover, participants from Riyadh exhibited a greater prevalence of cardiovascular risk factors compared to those in the Western Province ($p = 0.003$; Fig 2-3). However, no significant differences were found in the remaining meals consumed (Figs 2-1 and 2-2).

Significant variations were also observed in BMI, MET-minutes/week, and the intake of SF, high GL foods, CF, processed meat, and oily fish across different weighted age groups ($p = 0.002$ for oily fish; $p < 0.001$ for all other factors). Those aged 26 years or older exhibited greater corpulence than those aged 25 years or younger ($p = 0.004$ for those aged 20–25 years vs. those aged 26–30 years; $p < 0.001$ for the remaining age groups). Participants between the ages of 26 and 30 had a lower BMI than those between the ages of 35 and 40 ($p = 0.030$) and those between 46 and 50 ($p = 0.001$). Participants aged 46–50 had a significantly higher BMI than those aged 31–35 ($p = 0.022$) (Fig 3-1). In addition, participants aged 25 years or younger had lower MET values for PA than those aged 31 years or older ($p = 0.003$ for those aged 20–25 years vs. those aged 31–35 years; $p < 0.001$ for the other age ranges). Participants between the ages of 26 and 30 had lower MET values for PA than those aged 41 and over ($p = 0.002$ for those aged 41–45 years; $p = 0.024$ for those aged

**Table 2. Participants' BMI, MET, and eating patterns.**

|  | All (*N* = 968) | |
| --- | --- | --- |
| Variable | Mean | SD |
| Body Mass Index (kg/m²) | 23.31 | 5.18 |
| Metabolic Equivalents-minutes/week | 1397.10 | 946.86 |
| Saturated fat (g/day) | 38.71 | 25.77 |
| Glycemic Load (GL/day) | 177.74 | 131.79 |
| Cereal Fiber (g/day) | 18.35 | 19.61 |
| Processed Meat (g/day) | 18.14 | 60.82 |
| Oily Fish (servings/week) | 3.79 | 6.29 |
| Nuts (servings/day) | 0.57 | 0.98 |
| Overall Cardiovascular Disease Risk Factors | 4.98 | 1.54 |

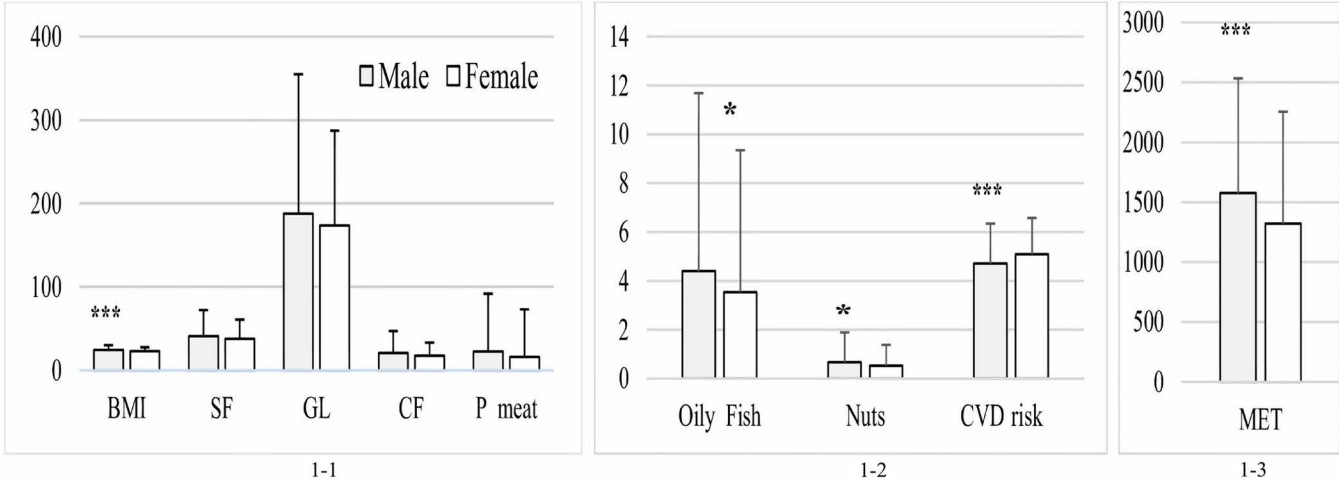

**Fig 1. Body mass index (BMI), metabolic equivalents (MET), food intake amount, and CVD risk among participants by gender.** (1-1) Dietary intake components: saturated fat (SF; g/day), glycemic load (GL), cereal fiber (CF; g/day), processed meat (P meat; g/day), and BMI. (1-2) Consumption of oily fish (servings/week) and nuts (servings/day), and CVD risk. (1-3) Physical activity levels measured in MET (min/week). Data are presented as mean ± standard deviation. *p < 0.05 and ***p < 0.001 indicate significant gender differences.

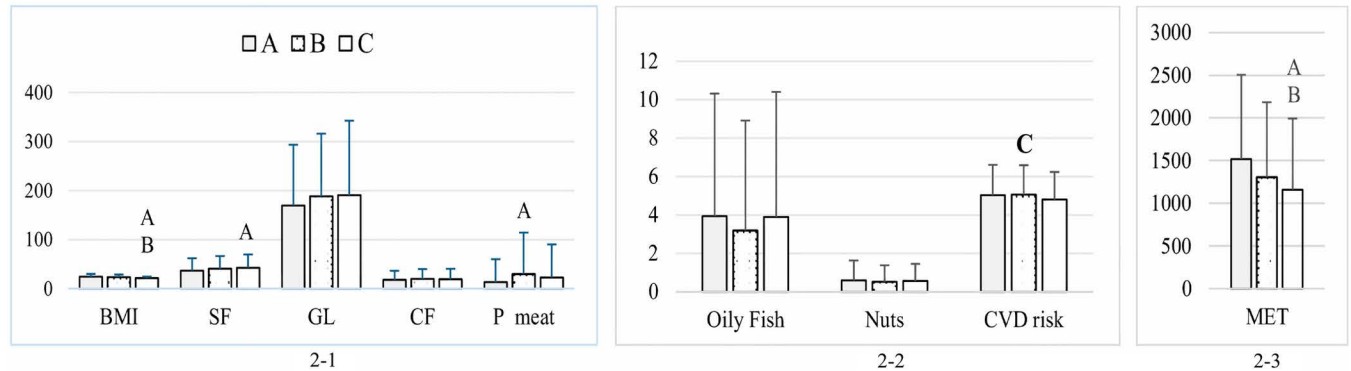

**Fig 2. BMI, MET, food intake, and CVD risk among participants by geographic region.** (2-1) Dietary intake components: saturated fat (SF; g/day), glycemic load (GL), cereal fiber (CF; g/day), processed meat (P meat; g/day), and BMI. (2-2) Consumption of oily fish (servings/week) and nuts (servings/day), and CVD risk. (2-3) Physical activity levels measured in MET (min/week). Different letters indicate significant differences between regions: A, differed significantly from Eastern Province; B, differed significantly from Riyadh Province; C, differed significantly from Western Province. Data are presented as mean ± standard deviation.

46–50 years) (Fig 3-3). In contrast, the dietary selections of the youngest participants exhibited elevated levels of SF, GL, and CF compared to participants aged 31 years and older (p = 0.003 for SF, 0.005 for GL, and 0.007 for CF relative to those aged 36–40 years; p < 0.001 for all other comparisons). Students aged 25 and younger showed lower processed meat consumption compared to those aged 36 and older (p = 0.03 for under 20 vs. 36–40 years; p = 0.15 for 20–25 vs. 45–50 years; p < 0.001 for all other comparisons). Additionally, students aged 20–25 consumed significantly less oily fish compared to those aged 46–50 (p = 0.014). No notable disparities were observed in the remaining factors, including the consumption of nuts or CVD risk factors across age groups (Fig 3-2).

The weighted participants' relationship status substantially affected their BMI, PA energy expenditure, SF intake, GL of foods ingested, and CF and nut intake. The p-values were all significant (<0.001, <0.001, 0.043, 0.003, 0.001, and 0.005,

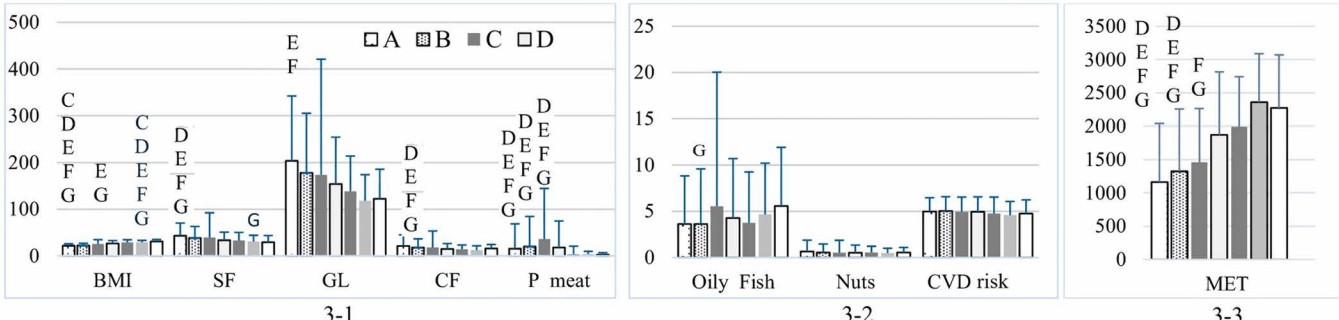

**Fig 3. BMI, MET, overall CVD risk, and food intake among participants by age.** (3-1) Dietary intake components: saturated fat (SF; g/day), glycemic load (GL), cereal fiber (CF; g/day), processed meat (P meat; g/day), and BMI. (3-2) Consumption of oily fish (servings/week) and nuts (servings/day), and CVD risk. (3-3) Physical activity levels measured in MET (min/week). Different letters indicate significant differences between age groups: C (26-30 years), D (31-35 years), E (36-40 years), F (41-45 years), and G (46-50 years), with each letter appearing only on the bar of the group that differs significantly. Data are presented as mean ± standard deviation.

respectively). Singles had a significantly lower BMI than those who were married or divorced ($p < 0.001$ and 0.010, respectively; Fig 4-1). Unmarried individuals also had lower MET values for PA (1285.74 ± 922.88 vs. 1840.09 ± 944.54; $p < 0.001$; Fig 4-3) but a greater consumption of foods with a high GL (184.2 ± 139.87 vs. 155.3 ± 90.89; $p = 0.039$) than married individuals. People who were widowed consumed significantly higher quantities of CF and nuts compared to single, married, and divorced individuals ($p = 0.005$ for both comparisons with single participants, $p = 0.001$ and 0.002 for comparisons with married participants, and $p = 0.002$ and 0.021 for comparisons with divorced participants; Figs 4-1 and 4-2).

Notable differences in the consumption of SF ($p = 0.009$), processed meat ($p = 0.001$), and oily fish ($p = 0.022$) were noted depending on the monthly weighted income. Participants with a monthly income of 3000 SAR or less had a higher consumption of SF than group B (3001–8000 SAR, $p = 0.005$), more processed meat than groups B ($p = 0.001$), C (8001–13,000 SAR, $p = 0.013$), D (13,001–18,000 SAR, $p = 0.002$), and E (18,001–23,000 SAR, $p = 0.028$), and more oily fish than group B ($p = 0.031$). No significant variations were observed in BMI, or in GL and CF intake (Fig 5-1); CVD risk factors or nut consumption (Fig 5-2); or MET (Fig 5-3).

## Prevalence of CVD risk

The incidence of CVD risk factors among individuals was 28.8% for overweight/obesity (males = 38.1%; females = 24.9%; $p < 0.001$), 9.4% for current smoking (males = 23.1%; females = 3.7%; $p < 0.001$), and 24.4% for physical inactivity (males = 17.8%; females = 27.1%; $p = 0.002$). The prevalence of CVD risk related to dietary habits was as follows: 69.5% for SF intake (males = 61.2%; females = 73%; $p < 0.001$), 58.8% for GL (males = 38.5%; females = 67.3%; $p < 0.001$), 16.1% for CF content (males = 21.7%; females = 13.8%; $p = 0.001$), 6.8% for processed meat consumption (males = 8.4%; females = 6.2%; $p = 0.209$), 72.4% for oily fish intake (males = 69.6%; females = 73.6%; $p < 0.001$), and 86.5% for nut consumption (males = 83.9%; females = 87.5%; $p = 0.133$) (Table 3). Among the participants, only 62 individuals (6.4%) were classified as having a reduced risk of CVD, with less than three risk factors. Of the remaining 906 persons (93.6%; males = 90.9%; females = 94.7%; $p = 0.004$), 62.3% (95% CI = 59.2%–65.3%) were classified as medium risk and 31.3% (95% CI = 28.4%–34.3%) were classified as high risk.

## Regressions

A stepwise logistic regression was used to investigate the effect of the demographic variables, and the risk factors associated with body composition, PA, and food intake on the likelihood of developing cardiovascular concerns. Two models were used to avoid confounding effects. Demographics and the total number of CVD risk factors were used in the first model. On

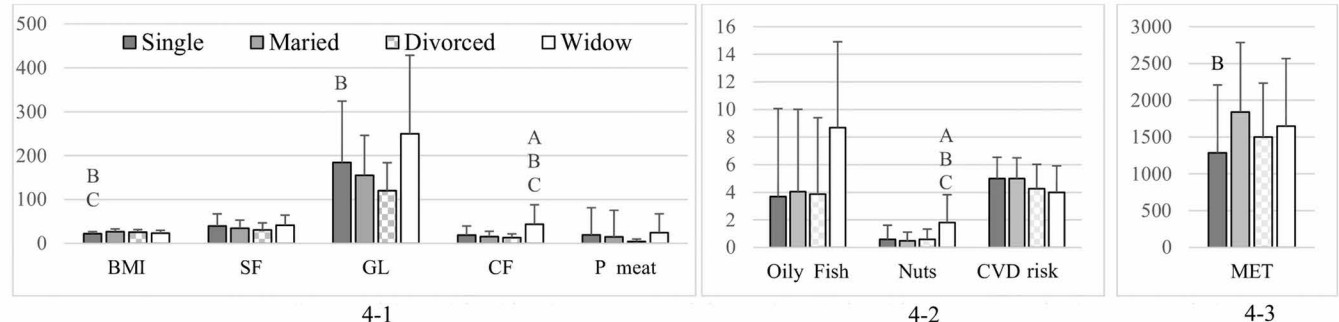

**Fig 4. BMI, MET, overall CVD risk, and food intake among participants by relationship status.** (4-1) Dietary intake components: saturated fat (SF; g/day), glycemic load (GL), cereal fiber (CF; g/day), processed meat (P meat; g/day), and BMI. (4-2) Consumption of oily fish (servings/week) and nuts (servings/day), and CVD risk. (4-3) Physical activity levels measured in MET (min/week). Different letters indicate significant differences between relationship status groups: A, significantly different from single participants; B, significantly different from married participants; and C, significantly different from divorced participants. Data are presented as mean±standard deviation.

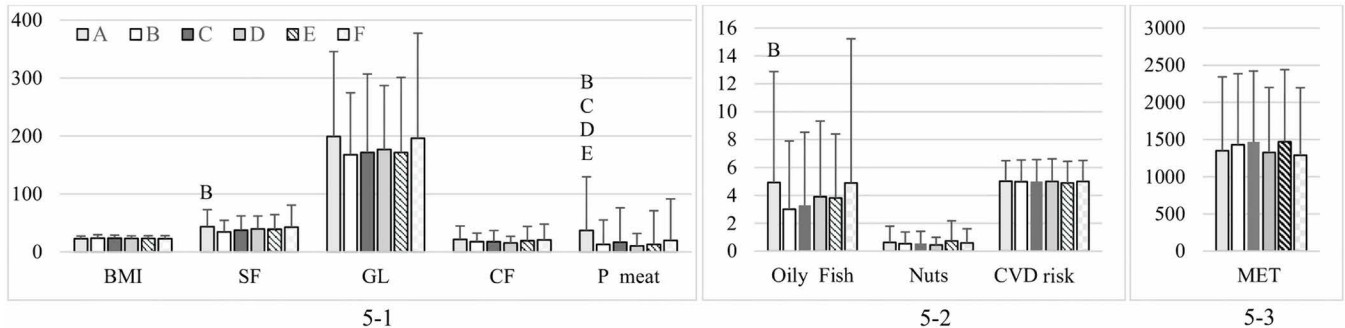

**Fig 5. BMI, MET, overall CVD risk, and food intake among participants by income.** (5-1) Dietary intake components: saturated fat (SF; g/day), glycemic load (GL), cereal fiber (CF; g/day), processed meat (P meat; g/day), and BMI. (5-2) Consumption of oily fish (servings/week) and nuts (servings/day), and CVD risk. (5-3) Physical activity levels measured in MET (min/week). Different letters indicate significant differences between income groups: B (3,001–8,000 Saudi Riyals [SAR]), C (8,001–13,000 SAR), D (13,001–18,000 SAR), and E (18,001–23,000 SAR), with each letter appearing only on the bar of the group that differs significantly. Data are presented as mean±standard deviation.

the other hand, demographics and the specific risk factors related to BMI, current smoking, PA level, and the consumption of SF, foods with a high GL, and intake of CF, processed meat, oily fish, and nuts were used in the second model.

Both models were significant ($\chi^2(9) = 75.63$, $p<0.001$ for the first model; $\chi^2(18) = 87.58$, $p<0.001$ for the second model) and explained 27.2% and 31.8% of the variance, respectively (Nagelkerke $R^2=0.272$ and 0.31, respectively). Both models indicated that being from Riyadh Province significantly decreased the likelihood of developing cardiovascular concerns by approximately 12% ($\chi^2(1) = 4.1$, $p=0.042$, Exp(β) = 0.120 for the first model; $\chi^2(1) = 3.88$, $p=0.049$, Exp(β) = 0.118 for the second model). Age also significantly affected the likelihood of developing cardiovascular issues ($\chi^2(6) = 24$, $p=0.001$ for the first model; $\chi^2(6) = 13$, $p=0.043$ for the second model); in particular, being aged 46–50 years increased the likelihood of developing cardiovascular concerns 5.14–8.93-fold.

Additionally, the first model revealed that each participant increased their likelihood of developing a cardiovascular problem ($\chi^2(1) = 34.3$, $p<0.001$) by 2.19-fold for each additional risk factor (**Table 4**). According to the second model, the risk factors most likely to influence the development of cardiovascular concerns were a high BMI, current smoking, insufficient PA, high SF intake, high GL, and inadequate CF and oily fish consumption. The Wald ratios for BMI (>25 kg/m²),

Table 3. Prevalence of CVD risk among the participants.

| Variable | Risk | N | All (N=968) | | | Males (N = 286) | | | Females (N = 682) | | | p |
|---|---|---|---|---|---|---|---|---|---|---|---|---|
| | | | Prevalence of risk (%) | 95% CI | | Prevalence of risk (%) | 95% CI | | Prevalence of risk (%) | 95% CI | | |
| | | | | Upper | Lower | | Upper | Lower | Upper | Lower | | |
| BMI | Low | 689 | 28.8 | 0.260 | 0.317 | 38.1 | 0.326 | 0.438 | 24.9 | 0.218 | 0.283 | <0.001*** |
| | High | 279 | | | | | | | | | | |
| Smoking | Low | 877 | 9.4 | 0.077 | 0.113 | 23.1 | 0.184 | 0.282 | 3.7 | 0.024 | 0.053 | <0.001*** |
| | High | 91 | | | | | | | | | | |
| Physical activity | Low | 732 | 24.4 | 0.217 | 0.272 | 17.8 | 0.137 | 0.225 | 27.1 | 0.239 | 0.305 | 0.002** |
| | High | 236 | | | | | | | | | | |
| Saturated fat | Low | 295 | 69.5 | 0.666 | 0.724 | 61.2 | 0.555 | 0.667 | 73.0 | 0.696 | 0.763 | <0.001*** |
| | High | 673 | | | | | | | | | | |
| Glycemic load | Low | 399 | 58.8 | 0.557 | 0.619 | 38.5 | 0.329 | 0.442 | 67.3 | 0.637 | 0.708 | <0.001*** |
| | Medium | 336 | | | | | | | | | | |
| | High | 233 | | | | | | | | | | |
| Cereal fiber | Low | 812 | 16.1 | 0.139 | 0.185 | 21.7 | 0.172 | 0.267 | 13.8 | 0.113 | 0.165 | 0.001*** |
| | Medium | 115 | | | | | | | | | | |
| | High | 41 | | | | | | | | | | |
| Processed meat | Low | 902 | 6.8 | 0.053 | 0.085 | 8.4 | 0.055 | 0.120 | 6.2 | 0.045 | 0.081 | 0.209 |
| | High | 66 | | | | | | | | | | |
| Oily fish | Low | 267 | 72.4 | 0.695 | 0.752 | 69.6 | 0.641 | 0.747 | 73.6 | 0.702 | 0.768 | <0.001*** |
| | Medium | 125 | | | | | | | | | | |
| | High | 576 | | | | | | | | | | |
| Nuts | Low | 131 | 86.5 | 0.842 | 0.885 | 83.9 | 0.794 | 0.879 | 87.5 | 0.849 | 0.899 | 0.133 |
| | High | 837 | | | | | | | | | | |
| Overall risk | Low | 62 | 93.6 | 0.919 | 0.950 | 90.9 | 0.872 | 0.939 | 94.7 | 0.929 | 0.962 | 0.004** |
| | Medium | 603 | | | | | | | | | | |
| | High | 303 | | | | | | | | | | |

**Note:** CI = confidence interval; *p < 0.05, **p < 0.01, ***p < 0.001.

current smoking (smoker), PA (inactive), SF (≥24.4 g/day for women; ≥ 30.5 g/day for men), GL (≥250/day for men; ≥ 206/day for women), CF (≤2.3 g/day for men; ≤ 2.5 g/day for women), and oily fish (less than 2 servings per week) and their corresponding p-values were $\chi^2(1) = 3.735$, $p = 0.054$; $\chi^2(1) = 13.56$, $p < 0.001$; $\chi^2(1) = 13.62$, $p < 0.001$; $\chi^2(1) = 9.11$, $p = 0.003$; $\chi^2(1) = 4.96$, $p = 0.026$; $\chi^2(1) = 7.76$, $p = 0.005$; $\chi^2(1) = 10.77$, $p = 0.001$; and $\chi^2(1) = 5.26$, $p = 0.022$, respectively. For these predictors, the probability of developing cardiovascular conditions increased by 11.70-, 6.54-, 7.61-, 4.79-, 7.00-, 24.58-, and 2.99-fold, respectively (**Table 4**).

## Discussion

This study aimed to determine the prevalence of CVD risk factors associated with body composition, PA, current smoking, and dietary behaviors among students in Saudi Arabia. A comprehensive analysis was performed to investigate variations according to geographic region, age, gender, relationship status, and monthly income of the participants. To mitigate non-response bias, demographic parameters were weighted to improve the repeatability of our findings on CVD risk factors, ensuring that our sample accurately represents the demographic profile of the target population and providing a more precise picture of the factors contributing to cardiovascular disorders.

**Table 4. Logistic Regression Models Predicting the Likelihood of Cardiovascular Issues Among Participants.**

Variables in the equation

| Variables in the Equation | Coefficient (B) | Wald | df | Sig. | Exp(β) | 95% CI for Exp(β) | |
|---|---|---|---|---|---|---|---|
| | | | | | | Lower | Higher |
| **Model 1** | | | | | | | |
| Region | | 5.1 | 2 | 0.077 | | | |
| Riyadh Province | -2.12 | 4.1 | 1 | 0.042* | 0.12 | 0.02 | 0.93 |
| Age | | 24.0 | 6 | <0.001*** | | | |
| Age (41–45 years old) | 1.76 | 5.7 | 1 | 0.016* | 5.81 | 1.38 | 24.45 |
| Age (46–50 years old) | 2.19 | 8.1 | 1 | 0.004** | 8.93 | 1.98 | 40.39 |
| Total CVD risks | 0.95 | 34.3 | 1 | <0.001*** | 2.60 | 1.89 | 3.57 |
| Constant | -8.07 | 54.0 | 1 | <0.001*** | 0.00 | | |
| **Model 2** | | | | | | | |
| Region | | 4.77 | 2 | 0.092 | | | |
| Riyadh Province | -2.14 | 3.88 | 1 | 0.049* | 0.118 | 0.01 | 0.99 |
| Age | | 13.00 | 6 | 0.043* | | | |
| Age (46–50 years old) | 1.64 | 3.73 | 1 | 0.050* | 5.14 | 0.98 | 27.13 |
| BMI (≥25 kg/m²) | 2.46 | 13.56 | 1 | <0.001*** | 11.70 | 3.16 | 43.34 |
| Smoking habits (smoker) | 1.88 | 13.62 | 1 | <0.001*** | 6.54 | 2.41 | 17.74 |
| PA level (inactive) | 2.03 | 9.11 | 1 | 0.003** | 7.61 | 2.04 | 28.45 |
| SF (≥24.4 g/day for women; ≥30.5 g/day for men) | 1.57 | 4.96 | 1 | 0.026* | 4.79 | 1.21 | 18.98 |
| GL | | 9.19 | 2 | 0.010** | | | |
| GL (≥250/day for men; ≥206/day for women) | 1.95 | 7.76 | 1 | 0.005** | 7.00 | 1.78 | 27.49 |
| CF | | 11.36 | 2 | 0.003** | | | |
| CF (≤2.3 g/day for men; ≤2.5 g/day for women) | 3.20 | 10.77 | 1 | 0.001*** | 24.58 | 3.63 | 166.40 |
| Oily fish | | 6.45 | 2 | 0.040* | | | |
| Oily fish (less than 2 servings per week) | 1.10 | 5.26 | 1 | 0.022* | 2.99 | 1.17 | 7.64 |
| Constant | -8.01 | 42.90 | 1 | <0.001*** | 0.00 | | |

**Note:** β = regression coefficients, df = degrees of freedom, Sig. = significance, Exp(β) = odds ratio, and CI = confidence interval; *p < 0.05, **p < 0.01, ***p < 0.001.

Multiple risk variables were identified as being associated with an increased risk of CVD. These included higher BMI, lower levels of PA, and dietary patterns characterized by a high intake of SF and high-GL meals, as well as a low consumption of essential preventive items such as CF, oily fish, and nuts. Demographic factors influenced risk, with older age (46–50 years), lower income (<3000 SAR), and marital status (e.g., married individuals) associated with increased vulnerability. Current smoking markedly elevated the risk of CVD, with regional variations indicating that individuals from Riyadh province exhibited a reduced probability of developing CVD in comparison to those from other provinces. Notably, gender differences revealed that men tended to have higher BMI and PA levels but exhibited a lower cumulative CVD risk compared to women, who displayed higher levels of dietary and physical inactivity-related risk factors.

The prevalence of CVD risk factors among Saudi participants is alarmingly high. A staggering 93.6% (95% CI = 91.9%–95%) of participants were found to be at risk for CVD, with 62.3% (95% CI = 59.2%–65.3%) categorized as medium risk and 31.3% (95% CI = 28.4%–34.3%) as high risk. Saudi females exhibited a greater risk of CVD than males, with 94.7% (p = 0.004) of female respondents presenting three or more CVD risk factors. The prevalent risk factors for Saudi males stemmed from a lack of protective factors, including insufficient nut consumption (83.9%) and minimal oily fish intake (69.6%). In contrast, they exhibited a heightened prevalence of factors that could elevate the risk of CVD, such as being

overweight/obesity (38.1%), current smoking (23.1%), and high SF consumption (61.2%). For females, prevalent variables comprised an excessive consumption of SF (73%), a high intake of foods with high GL (67.3%), and insufficient consumption of oily fish (73.6%) and nuts (87.5%). Nevertheless, both genders had a high intake of CF and a low consumption of processed meat, which represent positive factors. As a result, the risk values were 16.1% and 6.8% respectively for all participants, with males having a significantly higher CF intake (21.7%) than females (13.8%).

However, we found a lower prevalence of overweight and obesity compared to 2015 (48.4%) and 2019 (46.7%) [28,49] and compared to other Middle Eastern countries such as the UAE [50] and Kuwait [51], although it was higher than that of the UK [28]. A recent large survey collected data from all regions of Saudi Arabia and revealed a national weighted prevalence of obesity of 24.7% and a prevalence in the sample (unweighted) of 21.7% [52].

Several Saudi studies have reported a varying prevalence of obesity and overweight [38,52,53]. Thus, the actual prevalence of obesity in Saudi Arabia may be higher than the value reported in this study. Although there is scarce data to substantiate the decline in overweight and obesity, a potential explanation may be the youth of the participants in our study (86.1% were aged ≤ 30 years) and the fact they were predominantly unmarried and highly educated. Prior research indicates that the prevalence of obesity and overweight increases with age [28,38,53], lower educational attainment [54], and married status [55]. Furthermore, Althumiri et al. [52] argued that laws implemented in Saudi Arabia over the past decade may facilitate a reduction in obesity rates. Key changes between 2017 and 2020 included the introduction of PA classes in women's schools, the enactment of legislation allowing women to open fitness centers, the launch of quality-of-life programs as part of the Vision 2030 plan to encourage people to exercise, the implementation of a law mandating calorie information on restaurant menus in early 2019 (which was accompanied by extensive awareness campaigns in the Kingdom), and the introduction of an excise tax of 50% on sugary beverages and soft drinks and 100% on energy drinks in early 2019.

Variability in research outcomes can also be observed in participants' PA levels, current smoking, and dietary behaviors that elevate the CVD risk. Compared to the findings of Faleh [28], our participants exhibited a reduced prevalence of smoking, maintained an equivalent level of inactivity, and showed higher consumption of foods with elevated GL and an increased intake of CF, SF, processed meat, oily fish, and nuts. Al Moraie [38] reported a 71% prevalence of inactivity in a sample of 229 Saudis, with 62% of them being in Jeddah and 76% in Western Province. In a global cohort study of 2047 adults, Alhabib et al. [56] observed that 69.4% exhibited inadequate PA levels and 12.2% were smokers. The prevalence of current tobacco use rose from 12.2% in 2013 [57] to 21.4% in 2018 [58] and then decreased to 19.8% in 2019 [59]. In a prospective cohort study conducted in England with a 12-year follow-up, Jackson et al. [60] found that both former and current smokers had a greater risk of developing coronary heart disease (CHD) than non-smokers in unadjusted models (relative risk range (RR) of 1.20–2.34 for former smokers and 1.45–6.28 for current smokers). The risk of stroke was markedly elevated among present smokers compared to non-smokers (RR = 1.58). After adjusting for age, sex, ethnicity, affluence, alcohol use, BMI, and PA, the risk of developing CHD and stroke remained significant among current smokers (RR adjusted range = 1.55–1.93) compared to non-smokers. In addition, in unadjusted models, participants with low levels of PA had much greater chances of developing CHD and stroke than those with high levels of PA (specific risk ratio = 1.19–2.67). However, after accounting for age, sex, ethnicity, wealth, alcohol consumption, BMI, and smoking status, the likelihood of developing CHD and stroke decreased and became insignificant (RR adjusted range = 1.06–1.40).

Furthermore, our findings indicate that the CVD risk associated with SF intake and GL is high. Females exhibited a higher risk (SF = 73%; GL = 67.3%) than males (SF = 61.2%; GL = 38.5%), highlighting gender-specific eating habits that may predominantly affect SF intake and glycemic control in females. A high GL [10] and increased SF consumption [61] are well-established risk factors for CVD, indicating that dietary interventions to lower GL and SF intake may be advantageous, especially for women.

The participants' eating habits were also marked by high consumption of CF, predominantly among women, a minimal intake of processed meat in both genders, and low consumption of oily fish, primarily among males. Furthermore, most

participants did not consume nuts daily. Slavin [62] emphasized the preventive benefits of adequate CF consumption. A high CF intake correlates with a diminished risk of CVD, and our results underscore the necessity of sustaining elevated CF consumption, particularly among men, who ingest lower amounts of CF. Minimizing the intake of processed meat is crucial for managing CVD risk, according to the proven correlation between processed meat consumption and a heightened risk of CVD [63]. The diminished intake of oily fish and nuts, particularly among women, indicates that all participants ought to adhere to the advised consumption levels of these foods, recognized for their benefits on cardiovascular health, attributable to omega-3 fatty acids in oily fish [64–65] and healthy fats, fiber, and micronutrients in nuts [66]. This beneficial dietary practice represents a form of primary and secondary prevention of CVD [67].

This study aimed to assess the effects of demographic factors and CVD risk factors on the occurrence of cardiovascular concerns among participants. Both models indicated that the geographical region significantly affected the occurrence of CVDs among Saudi students. The stepwise logistic regression indicated that being from Riyadh Province reduced the probability of developing cardiovascular conditions by approximately 12%. Alqahtani and Alenazi [8] reported marked regional differences in the prevalence rates of diagnosed CVDs, with Western Province having the highest prevalence (1.9%), followed by Riyadh Province (1.7%). By contrast, Najran Province had the lowest prevalence of diagnosed CVD among Saudis (0.76%). Urbanization, lifestyle factors, socioeconomic factors, access to healthcare, and environmental factors contribute to the variability of CVDs among geographical regions. Thus, addressing these regional disparities through targeted public health interventions, increased access to healthcare, and educational campaigns may help lower the burden of CVDs in Saudi Arabia [69].

Our results, in accordance with several studies, demonstrated the significant effect of age on the prevalence of CVDs [70–71]. The prevalence of hypertension, ischemic heart disease, atrial fibrillation, and heart failure increases as the population ages [69]. Tash and Al-Bawardy [68] demonstrated that the prevalence of CVD increases gradually until the age of 50, at which point, it rises sharply. However, contrary to our results, where participants aged 46–50 were 8.93 times more likely to develop cardiovascular conditions, these authors found that the oldest group (≥ 65 years) had the highest CVD prevalence (11%), followed by those aged 60–64 years (6.5%), and the lowest prevalence corresponded to those under 40 years (1.2%).

In addition, the first model demonstrated that each participant may increase their likelihood of developing cardiovascular conditions by 2.19-fold for each additional risk factor. The risk factors most likely to influence the development of these diseases were the BMI (Exp(β) = 11.70), current smoking (Exp(β) = 6.54), PA (Exp(β) = 7.61), SF intake (Exp(β) = 4.79), GL (Exp(β) = 7.00), CF intake (Exp(β) = 24.58), and oily fish consumption (Exp(β) = 2.99). Notably, low CF intake and high BMI emerged as the most significant risk factors affecting the development of cardiovascular problems. However, the consumption of SF and oily fish had the lowest influence.

Consistent with our findings, the World Health Organization [2] affirms that a poor diet, insufficient PA, and tobacco smoking are the most important behavioral risk factors for CVDs. Studies have demonstrated that middle-aged adults who eat more whole grains have a lower BMI and less central obesity [72]. They also gain less weight over time compared to those who eat mostly refined grains [73]. Thus, CF, rather than vegetable or fruit fiber, appears to be more protective against metabolic syndrome and CVD development [74–75]. Dietary fiber, primarily sourced from cereals, may stimulate the secretion of gut hormones that regulate food consumption [62]. An experiment by Shivakoti et al. [76] looked at 4125 adults aged 65 and up. Shivakoti et al. [76] linked a 5 g/day increase in total fiber intake to significantly lower levels of C-reactive protein and interleukin-1 receptor antagonist and higher levels of soluble CD163. Among fiber sources, only CF was consistently associated with lower inflammation. Similarly, CF intake was associated with a lower CVD incidence (adjusted hazard ratio = 0.90; 95% CI = 0.81–1.00; 1941 incident cases). Inflammatory markers accounted for 1.5% of the link between CF and CVD, 14.2% for C-reactive protein, and 16.1% for their main component [37]. Recently, Mazur et al. [77] recommended that to reduce cardiovascular risk factors, individuals should restrict their consumption of SF to below 10% of the total caloric intake in instances of obesity, and to below 7% in cases of hypercholesterolemia, while augmenting

their intake of mono- and polyunsaturated fatty acids. The daily consumption of dietary fiber should range from 25 to 40 g. At least 200 g of vegetables and fruits should be consumed daily. Protein is crucial for tissue development and regeneration. Carbohydrates serve as the primary energy source; nonetheless, foods with a low glycemic index should be chosen. Moreover, dietary antioxidants, including fatty fish and nuts, combat free radicals and avert cellular harm [77].

This study's strengths include data from a substantial cohort of well-characterized students, encompassing comprehensive information on body composition, smoking behaviors, PA, dietary intake, and the incidence of cardiovascular conditions. The goal was to facilitate primordial prevention, which the American Heart Association considers the basis of optimal heart health [78]. Primordial prevention is most effective when initiated early, preferably during childhood. The current investigation also validates previously identified associations between BMI, smoking, PA, meals' GL, and the intake of SF, CF, and fatty fish, in relation to the incidence of cardiovascular issues.

Nonetheless, the study's limitations encompass the possibility of non-response bias due to a significant number of non-responding participants, as individuals who choose not to participate may systematically differ from those who did, affecting the generalizability of the results. Disparities in demographics such as geography, gender, age, social relationship, and income between respondents and non-respondents may result in bias, especially in evaluating CVD risk factors. Despite the implementation of various strategies to alleviate this bias — including random stratified sampling, motivational cover letter, explicit survey instructions, in-person reminders from instructors, assurances of confidentiality, flexible scheduling, data collection across two waves, and demographic weighting — there remains potential for unmeasured variations from non-respondents to affect the study results.

Additional limitations include the lack of data on supplementary risk factors, such as stress, air pollution, and salt intake, which could influence the observed correlations. Measurement inaccuracies linked to consumption data from food frequency questionnaires also pose a limitation, as self-reported data is subject to recall bias and social desirability bias. Furthermore, the small male sample size limits the generalizability of the results for this subgroup. The use of self-reported data for the diagnosis of cardiovascular conditions, as well as for evaluating body composition and PA, may also limit the study. These auto-declared checks may exclude individuals who are unaware of their medical diagnoses or who may overestimate specific behaviors and underestimate food intake [79]. Subsequent investigations ought to assess these factors through direct measurement and by employing diagnostic instruments and reference metrics.

## Conclusions

This study revealed a concerning prevalence of CVD risk factors among Saudi students, as 93.6% of the participants were at risk for CVD. After applying gender weighting, females demonstrated a higher risk than males, with 94.7% exhibiting three or more CVD risk factors. For Saudi males, predominant risk factors arose from a deficiency of preventive elements, notably inadequate nut consumption (83.9%) and limited oily fish intake (69.6%). High rates of overweight/obesity (38.1%), current smoking (23.1%), and excessive SF consumption (61.2%) also contributed to the CVD risk. For females, significant risk factors included excessive SF consumption (73%), high intake of foods with elevated GL (67.3%), and inadequate consumption of oily fish (73.6%) and nuts (87.5%). However, both genders demonstrated some favorable eating practices, such as a high consumption of CF and a low intake of processed meats. The stepwise logistic regression analysis revealed that geographical region significantly influenced the prevalence of CVDs among participants, with residency in Riyadh Province decreasing the likelihood of developing cardiovascular conditions by approximately 12%. Age also significantly influenced the prevalence of these disorders, with those aged 46–50 years exhibiting a 5.14- to 8.93-fold increase in the chance of developing cardiovascular issues. Furthermore, every additional risk factor elevated the likelihood of developing such abnormalities by 2.19-fold. The primary risk variables likely to affect the onset of these diseases were BMI, current smoking, insufficient PA, high SF intake, high GL, and inadequate CF and oily fish consumption. Low CF consumption and high BMI were identified as the most critical risk factors influencing the onset of cardiovascular issues, while the intake of SF and oily fish had a comparatively lower influence.

## Acknowledgments

The authors wish to thank all participants for their valuable contribution to this study.

## Author contributions

**Conceptualization:** Mohammed Shaab Alibrahim, Mohamed Ahmed Said, Abdulmalek K. Bursais.

**Data curation:** Mohammed Shaab Alibrahim, Mohamed Ahmed Said, Abdulmalek K. Bursais, Mohamed Abdelmoniem Abdelrahman, Hasnaa Hamdi Mohamed, Ahmad K. Hassan, Abdulrahman I Alaqil, Norah S. Almudaires, Narjis M.A. Alamer, Osama Eid Aljuhani, Hind Omer Salem Alshaghdali, Amani Hamzah ALjahani, Zuhair A. Al Salim, Atyh Abdullah Hadadi, Najeeb Abbas Aldarushi, Amal Nassir Alkuraieef, Ghareeb O. Alshuwaier.

**Formal analysis:** Mohamed Ahmed Said, Abdulmalek K. Bursais, Ibrahim I. Atta, Mohamed Abdelmoniem Abdelrahman, Atyh Abdullah Hadadi, Ghareeb O. Alshuwaier.

**Funding acquisition:** Mohammed Shaab Alibrahim.

**Investigation:** Mohammed Shaab Alibrahim, Mohamed Ahmed Said, Abdulmalek K. Bursais, Ibrahim I. Atta, Mohamed Abdelmoniem Abdelrahman, Hasnaa Hamdi Mohamed, Abdulrahman I Alaqil, Norah S. Almudaires, Narjis M.A. Alamer, Osama Eid Aljuhani, Hind Omer Salem Alshaghdali, Amani Hamzah ALjahani, Zuhair A. Al Salim, Najeeb Abbas Aldarushi, Amal Nassir Alkuraieef, Ghareeb O. Alshuwaier.

**Methodology:** Mohammed Shaab Alibrahim, Mohamed Ahmed Said, Abdulmalek K. Bursais, Ahmad K. Hassan, Osama Eid Aljuhani.

**Project administration:** Mohammed Shaab Alibrahim, Abdulmalek K. Bursais, Ibrahim I. Atta.

**Resources:** Abdulmalek K. Bursais, Mohamed Abdelmoniem Abdelrahman, Hasnaa Hamdi Mohamed, Ahmad K. Hassan, Abdulrahman I Alaqil, Osama Eid Aljuhani, Amani Hamzah ALjahani, Najeeb Abbas Aldarushi.

**Software:** Abdulmalek K. Bursais, Ibrahim I. Atta, Hasnaa Hamdi Mohamed, Norah S. Almudaires, Narjis M.A. Alamer, Atyh Abdullah Hadadi.

**Supervision:** Mohammed Shaab Alibrahim, Mohamed Ahmed Said.

**Validation:** Mohammed Shaab Alibrahim, Mohamed Ahmed Said, Abdulmalek K. Bursais, Ibrahim I. Atta, Mohamed Abdelmoniem Abdelrahman, Hasnaa Hamdi Mohamed, Ahmad K. Hassan, Abdulrahman I Alaqil, Norah S. Almudaires, Narjis M.A. Alamer, Osama Eid Aljuhani, Hind Omer Salem Alshaghdali, Amani Hamzah ALjahani, Zuhair A. Al Salim, Atyh Abdullah Hadadi, Najeeb Abbas Aldarushi, Amal Nassir Alkuraieef, Ghareeb O. Alshuwaier.

**Visualization:** Ibrahim I. Atta, Mohamed Abdelmoniem Abdelrahman, Hasnaa Hamdi Mohamed, Ahmad K. Hassan, Abdulrahman I Alaqil, Norah S. Almudaires, Narjis M.A. Alamer, Osama Eid Aljuhani, Hind Omer Salem Alshaghdali, Amani Hamzah ALjahani, Zuhair A. Al Salim, Atyh Abdullah Hadadi, Najeeb Abbas Aldarushi, Amal Nassir Alkuraieef, Ghareeb O. Alshuwaier.

**Writing – original draft:** Mohamed Ahmed Said.

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
