## [Decision Letter · Decision Letter 0]

22 Oct 2024

PONE-D-24-43695

Risk Factors for Cardiovascular Disease Among Saudi Students: Association with BMI, Current Smoking, Level of Physical Activity, and Dietary Habits

PLOS ONE

Dear Dr. Said,

Thank you for submitting your manuscript to PLOS ONE. After careful consideration, we have decided that your manuscript does not meet our criteria for publication and must therefore be rejected.

I am sorry that we cannot be more positive on this occasion, but hope that you appreciate the reasons for this decision.

Kind regards,

Erfan Taherifard, MD

Academic Editor

PLOS ONE

Reviewers' comments:

Reviewer's Responses to Questions

**Comments to the Author**

1. Is the manuscript technically sound, and do the data support the conclusions?

Reviewer #1: Yes

Reviewer #2: No

2. Has the statistical analysis been performed appropriately and rigorously? 

Reviewer #1: I Don't Know

Reviewer #2: Yes

3. Have the authors made all data underlying the findings in their manuscript fully available?

Reviewer #1: Yes

Reviewer #2: Yes

4. Is the manuscript presented in an intelligible fashion and written in standard English?

Reviewer #1: Yes

Reviewer #2: Yes

5. Review Comments to the Author

Reviewer #1: Dear Editorial team,

Thanks for the invitation to peer-review this manuscript. It was well written and the topic was really important. I think this manuscript could be published in the current format.

Reviewer #2: Dear editorial team,

Thank you for extending the invitation to peer review a manuscript for PLOS ONE. The manuscript addresses an important topic: the factors associated with cardiovascular diseases among students. However, there are several important issues within the manuscript that makes it unsuitable for publication.

1. The most important issue of this manuscript is that the low response rate to the invitations sent to the students 56.62%. The authors should provide their calculation regarding the minimum needed sample size in this study and then evaluate whether this high rate of non-response has affected their power of study or not. Besides, there might be non-response bias, meaning that those who didn't respond could differ systematically from those who did. This could affect the generalizability of the findings.

2. Besides, the fact that one-fourth of these respondents provided incomplete questionnaires, leading to their exclusion from the analysis, raises concerns about the credibility of the results.

3. Additionally, highlighting that the study focuses on students is inconsistent with the broad age range of 18-50 years used in the sample. This discrepancy is especially problematic because the study investigates cardiovascular diseases, which are more prevalent in older individuals. Including older participants could dilute the study’s focus on the student population, as cardiovascular risk factors vary significantly between younger and older age groups.

6. PLOS authors have the option to publish the peer review history of their article (what does this mean? ). If published, this will include your full peer review and any attached files.

**Do you want your identity to be public for this peer review?** For information about this choice, including consent withdrawal, please see our Privacy Policy .

Reviewer #1: No

Reviewer #2: No

- - - - -

---

## [Author Response · Author response to Decision Letter 0]

4 Nov 2024

PONE-D-24-43695

Risk Factors for Cardiovascular Disease Among Saudi Students: Association with BMI, Current Smoking, Level of Physical Activity, and Dietary Habits.

PLOS ONE

Dear editor,

Thank you for your informative and insightful comments. We appreciate the opportunity to address your issues and are grateful for your ideas, which have helped improve the quality of our manuscript. We have made a number of improvements, and as you will see, these changes have improved both our methodology and the clarity of our results.

Recognizing the potential influence of non-response bias on study validity, we included additional measures to mitigate this risk and enhance sample representativeness. Participants were chosen using stratified random sampling throughout three provinces—East, West, and Riyadh—while taking gender, geographic location, and academic discipline into account, as stated in the amended text. To achieve the desired sample size of 384 (based on a 95% confidence level and a 5% margin of error), an initial sample size of 1,920 was planned, with a 20% response rate due to the length of the questionnaire (90 items). Data collecting took place in two waves (March-April and November-December 2023), with enhanced response rates resulting from classroom visits and instructor encouragement.

In addition, we recalculated all statistics and included the changes throughout the manuscript. To increase transparency, the modified portions are marked in yellow. Post-stratification weights were used to align our sample's demographics with the larger student population, thereby lowering non-response bias. Demographic factors were included in our analysis to account for any residual variation. Moreover, despite these attempts, we recognize that unmeasured differences between respondents and non-respondents may influence our findings. As a result, we have added non-response bias as a study limitation at the end of the manuscript.

We hope that these clarifications satisfy your concerns. Please see below for full responses to each reviewer comment.

Thank you for your helpful feedback.

Sincerely,

Dr. Mohamed Ahmed SAID

Response point-by-point

Reviewer1:

Thanks for the invitation to peer-review this manuscript. It was well written, and the topic was really important. I think this manuscript could be published in the current format.

Response:

I am extremely grateful for your kind words and thoughtful criticism. We are delighted to hear that you found the manuscript well written and the subject interesting. We are deeply grateful for your support and favorable assessment of our work. We look forward to disseminating our results to a wider audience, thanks in part to your help.

Reviewer 2:

Thank you for extending the invitation to review a manuscript for PLOS ONE. The manuscript addresses an important topic: the factors associated with cardiovascular diseases among students. However, there are several important issues within the manuscript that make it unsuitable for publication.

Comment 1:

The most important issue of this manuscript is the low response rate to the invitations sent to the students (56.62%). The authors should provide their calculation regarding the minimum sample size needed in this study and then evaluate whether this high rate of non-response has affected their power of study or not. Besides, there might be non-response bias, meaning that those who didn't respond could differ systematically from those who did. This could affect the generalizability of the findings.

Response:

Thank you for this significant observation concerning response rate and the possibility of non-response bias. Please, note that the response rate of our study, 56.62%, exceeds the average response rate for online surveys, which is 44.1% (Wu et al., 2022).

In addition, and to guarantee the robustness of our findings, we conducted a power analysis before initiating the investigation, establishing a minimum sample size of 384 participants based on a 95% confidence level and projected margin of error. The total sample size was 968 participants, surpassing the minimum criteria in both data collection rounds, hence enhancing the statistical power and reliability of the study findings.

Nonetheless, as you indicated, non-response bias is a challenge in survey-based research, and we acknowledge that responders may possess distinct features compared to non-respondents, potentially affecting the generalizability of our findings. To mitigate this bias, we employed stratified random sampling to enhance representativeness and incorporated follow-up contact to diminish the probability of systematic disparities. Furthermore, we conducted a demographic analysis comparing the responder group with the university population data in Saudi Arabia, and the parallels in age and gender distribution indicate that our sample is adequately representative of the broader student population. Moreover, this point was added as a study limitation as the non-response bias remains possible.

We trust this resolves your worries and elucidates the actions implemented to mitigate non-response bias and uphold the integrity of our study.

Comment 2:

Besides, the fact that one-fourth of these respondents provided incomplete questionnaires, leading to their exclusion from the analysis, raises concerns about the credibility of the results.

Response:

Thank you for emphasizing this point. We acknowledge that the exclusion of some incomplete responses may cast doubt on the study's trustworthiness. We contend that this strategy was essential to guaranteeing the robustness and dependability of our findings.

The questionnaire employed in this study had 90 items aimed at thoroughly evaluating cardiovascular disease risk factors, activities, and demographic information across various dimensions of the respondents' lifestyles. Although this length probably led to a significant number of incomplete replies, we prioritized the collection of quality and reliable data over sheer numbers. Incomplete responses may have introduced inconsistencies or gaps that could undermine the validity of our results; however, by using solely complete responses, we ensured enhanced data quality and mitigated any biases arising from absent information.

To mitigate the potential stress of an extensive survey, we employed various methods to enhance students' desire to participate, including follow-up reminders. We acknowledge the constraints of partial responses and value your understanding of our choice to emphasize data completeness for thorough and dependable analysis.

Comment 3:

Additionally, highlighting that the study focuses on students is inconsistent with the broad age range of 18–50 years used in the sample. This discrepancy is especially problematic because the study investigates cardiovascular diseases, which are more prevalent in older individuals. Including older participants could dilute the study’s focus on the student population, as cardiovascular risk factors vary significantly between younger and older age groups.

Response:

Thank you for this perceptive and precise remark. We understand your concern regarding the broad age range of 18–50 years in a study focused on students.

The authors have intentionally extended the age range by including e-learning students who may differ in health status, particularly concerning non-communicable diseases, from traditional students. Cardiovascular disease risk factors related to lifestyle tend to accumulate over time, often becoming more evident with advancing age. By including a wider age range, we aimed to identify early lifestyle behaviors and risk factors in students that may have long-term consequences.

Even though the incidence of CVD rises with age, these conditions do not appear suddenly and develop over time in response to a person's lifestyle choices. This study sought to uncover lifestyle-related risks among younger students by observing patterns in older students as well. This approach allows for the early identification of CVD-related markers in younger students, supporting timely interventions and primary prevention efforts.

---

## [Decision Letter · Decision Letter 1]

5 Dec 2024

PONE-D-24-43695R1Risk Factors for Cardiovascular Disease Among Saudi Students: Association with BMI, Current Smoking, Level of Physical Activity, and Dietary HabitsPLOS ONE

Dear Dr. Said,

Thank you for submitting your manuscript to PLOS ONE. After careful consideration, we feel that it has merit but does not fully meet PLOS ONE’s publication criteria as it currently stands. Therefore, we invite you to submit a revised version of the manuscript that addresses the points raised during the review process.

We look forward to receiving your revised manuscript.

Kind regards,

Wenpeng You

Academic Editor

PLOS ONE

2. Thank you for stating the following financial disclosure: [The Deanship of Scientific Research, King Faisal University, Al-Ahsa 31982, Saudi Arabia, financed this study (INST037).]

3. We note that your Data Availability Statement is currently as follows: [All relevant data are within the manuscript and its supporting information files.]

The values behind the means, standard deviations and other measures reported;

The values used to build graphs;

The points extracted from images for analysis.

5. We note that your manuscript is not formatted using one of PLOS ONE’s accepted file types. Please reattach your manuscript as one of the following file types: .doc, .docx, .rtf, or .tex (accompanied by a .pdf).

If your submission was prepared in LaTex, please submit your manuscript file in PDF format and attach your .tex file as “other.”

6. Please ensure that you refer to Figure 1 in your text as, if accepted, production will need this reference to link the reader to the figure.

Additional Editor Comments (if provided):

Reviewers' comments:

Reviewer's Responses to Questions

**Comments to the Author**

1. If the authors have adequately addressed your comments raised in a previous round of review and you feel that this manuscript is now acceptable for publication, you may indicate that here to bypass the “Comments to the Author” section, enter your conflict of interest statement in the “Confidential to Editor” section, and submit your "Accept" recommendation.

Reviewer #3: All comments have been addressed

Reviewer #4: (No Response)

2. Is the manuscript technically sound, and do the data support the conclusions?

Reviewer #3: Yes

Reviewer #4: Yes

3. Has the statistical analysis been performed appropriately and rigorously? 

Reviewer #3: Yes

Reviewer #4: Yes

4. Have the authors made all data underlying the findings in their manuscript fully available?

Reviewer #3: Yes

Reviewer #4: Yes

5. Is the manuscript presented in an intelligible fashion and written in standard English?

Reviewer #3: Yes

Reviewer #4: Yes

6. Review Comments to the Author

Reviewer #3: It is an honour to review the manuscript. This is an interesting study on “Risk Factors for Cardiovascular Disease Among Saudi Students: Association with BMI, Current Smoking, Level of Physical Activity, and Dietary Habits”.

My comments are appended.

Abstract:

In my opinion background of the abstract need to be more specific based on research gap. Methods need to be rewritten to add details about metrics of data and data analysis.

Introduction:

In the first paragraph lots of unnecessary information. The author needs to be more specific to CVD risk and research gap in this field in Saudi Arabia. The whole introduction needs to be shortened to make it more focused.

Methods:

The inclusion criteria are quite broad. As the study focused on CVD risk factor in students, it is better to explain the age range or student of secondary or tertiary or both were considered in the inclusion criteria.

Results:

Results has been written in right direction.

Discussion:

In the first paragraph the author didn’t report the key findings (specific factors found to be associated) and then need to elaborate based on their findings.

General comments:

The manuscript is interesting and but need improvements in the introduction and discussion section.

Reviewer #4: An interested study investigates the CVD risk factors among Saudi using valid and suitable measures and appropriate sample size.

Abstract

Line 11 in the abstract, mention that "being between 46 and 50 years of age" but the study conducted on students. How could justify having this age group among students?

Line 15, Is oily fish consumption increased risk of CVD?

The introduction is well written and comprehensively addresses topics, but has some minor point need to be addressed.

First paragraph needs more reference, and it would be better to use article not website

The authors extensively using reference number 1

Prevalence for CVD among Saudi is needed

Line 24, I do not think the mentioning "excessive alcohol consumption" needed when you studied Saudi population.

Page 5 line 18, authors mentioned reference in two different styles {[24] (Aljefree and Ahmed, 2015).}. correct this

Methods

Page 6 line 14, age from18-50 Is not age of students, either change title and remove students or remove participants above age 35.

First authors mentioned the use for Saudi Food Frequency Questionnaire which include questions about demographics, smoking, physical activity, and food consumption and eating habits, but the mention that other questionnaires were used to measure demographics, smoking, physical activity, and food consumption and eating habits with citing other reference than reference 26 for SFFQ. This quite confusing Why two questionnaires to measure the same variable Why not use the best one only? This need justification.

Result well described data and discussion provide insightful interpretations.

7. PLOS authors have the option to publish the peer review history of their article (what does this mean? ). If published, this will include your full peer review and any attached files.

**Do you want your identity to be public for this peer review?** For information about this choice, including consent withdrawal, please see our Privacy Policy .

Reviewer #3: **Yes: ** Md Kamruzzaman

Reviewer #4: No

---

## [Author Response · Author response to Decision Letter 1]

16 Dec 2024

Author's point-by-point response

Reviewer #3: It is an honor to review the manuscript. This is an interesting study on “Risk Factors for Cardiovascular Disease Among Saudi Students: Association with BMI, Current Smoking, Level of Physical Activity, and Dietary Habits”.

My comments are appended.

Abstract:

In my opinion the background of the abstract needs to be more specific based on research gap. Methods need to be rewritten to add details about metrics of data and data analysis.

Response: The background and methods sections were revised, and the necessary information was incorporated.

Introduction:

In the first paragraph lots of unnecessary information. The author needs to be more specific to CVD risk and research gap in this field in Saudi Arabia. The whole introduction needs to be shortened to make it more focused.

Response: Several revisions have been made to the introduction to enhance clarity and relevance by focusing more specifically on CVD risks. Redundant information was removed, while essential details were retained to maintain the integrity of the content. Consequently, the word count was reduced from 1,147 to 841 words (after adding a new paragraph reporting the CVDs prevalence in Saudi Arabia suggested by Reviewer 2), improving brevity without compromising key information.

Methods:

The inclusion criteria are quite broad. As the study focused on CVD risk factor in students, it is better to explain the age range or student of secondary or tertiary or both were considered in the inclusion criteria.

Response: Only Saudi university students aged 18–50 years participated in this study. The research specifically targeted tertiary-level students across all academic stages, including undergraduate, postgraduate, and distance/online learning. Secondary school students were excluded from studying. The broad age range reflects the diverse demographic of Saudi university students, which includes younger traditional students and older individuals who may be pursuing postgraduate studies or returning to education later in life.

To enhance clarity, the text has been updated to explicitly state that only tertiary-level students were included and to provide a rationale for the specified age range. This explanation aligns the inclusion criteria with the study's focus on CVD risk factors within this cohort. (Page 6; Lines 16-18)

Results:

Results have been written in the right direction.

Response: Thank you very much for your encouraging comments. Your opinion means a great deal to us and inspires us to continually surpass ourselves in striving for new heights of performance.

Discussion:

In the first paragraph the author didn’t report the key findings (specific factors found to be associated) and then needed to elaborate based on their findings.

Response: The key findings have been added at the beginning of the Discussion section (Page 19; Lines 9-18)

General comments:

The manuscript is interesting and but need improvements in the introduction and discussion section.

Response: The authors would like to thank the reviewer for their insightful and constructive comments, which significantly contributed to the enhancement of the manuscript.

Reviewer #4: An interested study investigates the CVD risk factors among Saudi using valid and suitable measures and appropriate sample size.

Abstract

Line 11 in the abstract, mention that "being between 46 and 50 years of age" but the study conducted on students. How could justify having this age group among students?

Response: Many Saudi students return to university later in life due to socioeconomic issues including profession changes or returning to school after or while working. Our study focused on 18–30-year-olds, but we also recognized non-traditional students in continuing education and professional development programs, such as those in their late 20s to early 50s. This age group was included in our research to represent the overall population and assess cardiovascular disease risk at different life stages. This is explained in several manuscript parts.

Line 15, Is oily fish consumption increased risk of CVD?

Response: No, oily fish has a protective effect, as mentioned several times throughout the manuscript. However, the insufficient consumption of oily fish by participants represents a serious concern.

The introduction is well written and comprehensively addresses topics but has some minor points that need to be addressed.

First paragraph needs more reference, and it would be better to use article not website

The authors extensively using reference number 1.

Response: Modifications have been made to the introduction section to enhance clarity and increase the number of references. In the updated version, reference 1 appears only once. Furthermore, we have ensured that references are balanced and appropriately distributed. We have restricted website citations to credible and authoritative sources, including WHO, CDC, NHS, ADA, and pertinent ministries, to ensure the manuscript's quality and reliability.

Prevalence for CVD among Saudi is needed

Response: A new paragraph reporting the prevalence of CVD in Saudi Arabia have been added. (Pages 3-4; Lines 14-24 and 1-7)

Line 24, I do not think the mentioning "excessive alcohol consumption" needed when you studied Saudi population.

Response: This aspect was included in the overall discussion of behavioral risk factors for CVDs. While it may not relate directly to the Saudi population, it does provide a broader perspective for worldwide CVD risk.

Page 5 line 18, authors mentioned reference in two different styles {[24] (Aljefree and Ahmed, 2015).}. correct this

Response: Corrected.

Methods

Page 6 line 14, age from18-50 Is not age of students, either change title and remove students or remove participants above age 35.

Response: In Saudi Arabia, it is common for individuals to pursue higher education or professional studies later in life, often due to career transitions or opportunities to continue education through in-person, online, or continuing education programs. Therefore, the inclusion of participants between the ages of 35 and 50 is in line with the reality of Saudi Arabia's higher education demographics. In addition, this broader age range allows for a more comprehensive analysis of CV risk factors among participants. While we considered replacing the term "students" with "university attendees" for greater precision, this alternative could lead to confusion by including learners, academics, administrators, and staff. Therefore, we opted to retain the term "students" and provide a detailed explanation in the text to ensure clarity and consistency.

First authors mentioned the use for Saudi Food Frequency Questionnaire which include questions about demographics, smoking, physical activity, and food consumption and eating habits, but the mention that other questionnaires were used to measure demographics, smoking, physical activity, and food consumption and eating habits with citing other reference than reference 26 for SFFQ. This quite confusing Why two questionnaires to measure the same variable Why not use the best one only? This need justification.

Response: The Saudi Food Frequency Questionnaire (SFFQ) integrates items from other validated questionnaires in certain sections. For instance, the third section of the SFFQ, which addresses physical activity and sedentary behaviors, includes items from the Arabic version of the International Physical Activity Questionnaire (IPAQ). This point has been clearly addressed in the revised version of the manuscript, specifically highlighting that the third section of the SFFQ incorporates items from the validated Arabic version of the International Physical Activity Questionnaire (IPAQ) to assess physical activity and sedentary behaviors. (Page 8; Lines 1-6)

---

## [Decision Letter · Decision Letter 2]

3 Mar 2025

Risk Factors for Cardiovascular Disease Among Saudi Students: Association with BMI, Current Smoking, Level of Physical Activity, and Dietary Habits

PONE-D-24-43695R2

Dear Dr. Said,

We’re pleased to inform you that your manuscript has been judged scientifically suitable for publication and will be formally accepted for publication once it meets all outstanding technical requirements.

Kind regards,

Wenpeng You

Academic Editor

PLOS ONE

Additional Editor Comments (optional):

Reviewers' comments:

Reviewer's Responses to Questions

**Comments to the Author**

1. If the authors have adequately addressed your comments raised in a previous round of review and you feel that this manuscript is now acceptable for publication, you may indicate that here to bypass the “Comments to the Author” section, enter your conflict of interest statement in the “Confidential to Editor” section, and submit your "Accept" recommendation.

Reviewer #3: All comments have been addressed

Reviewer #4: (No Response)

2. Is the manuscript technically sound, and do the data support the conclusions?

Reviewer #3: Yes

Reviewer #4: Yes

3. Has the statistical analysis been performed appropriately and rigorously? 

Reviewer #3: Yes

Reviewer #4: Yes

4. Have the authors made all data underlying the findings in their manuscript fully available?

Reviewer #3: Yes

Reviewer #4: Yes

5. Is the manuscript presented in an intelligible fashion and written in standard English?

Reviewer #3: Yes

Reviewer #4: Yes

6. Review Comments to the Author

Reviewer #3: Thanks for addressing the issue raised during the review process. The quality of the manuscript has now been increased.

Reviewer #4: (No Response)

7. PLOS authors have the option to publish the peer review history of their article (what does this mean? ). If published, this will include your full peer review and any attached files.

**Do you want your identity to be public for this peer review?** For information about this choice, including consent withdrawal, please see our Privacy Policy .

Reviewer #3: **Yes: ** Md Kamruzzaman

Reviewer #4: No

---

## [Editor Report · Acceptance letter]

PONE-D-24-43695R2

PLOS ONE

Dear Dr. Said,

I'm pleased to inform you that your manuscript has been deemed suitable for publication in PLOS ONE. Congratulations! Your manuscript is now being handed over to our production team.

Kind regards,

on behalf of

Dr. Wenpeng You

Academic Editor

PLOS ONE